# *Pseudomonas* Infection Affects the Growth and Development of *Aphis gossypii* by Disrupting Energy Metabolism and Reproductive Processes

**DOI:** 10.3390/insects16030238

**Published:** 2025-02-24

**Authors:** Qiqing Yu, Ruichang Niu, Xueke Gao, Junyu Luo, Jinjie Cui, Li Wang, Xiangzhen Zhu

**Affiliations:** 1Zhengzhou Research Base, State Key Laboratory of Cotton Bio-Breeding and Integrated Utilization, Institute of Cotton Research, Chinese Academy of Agricultural Sciences, Anyang 455000, China; yuqq131222@163.com (Q.Y.); niurcyx@163.com (R.N.); 15036138389@163.com (X.G.); luojunyu1818@126.com (J.L.); cuijinjie@caas.cn (J.C.); 2Zhengzhou Research Base, State Key Laboratory of Cotton Bio-Breeding and Integrated Utilization, School of Agricultural Sciences, Zhengzhou University, Zhengzhou 450001, China; 3State Key Laboratory of Cotton Bio-Breeding and Integrated Utilization, Henan University, Kaifeng 475000, China

**Keywords:** *Pseudomonas*, *Aphis gossypii* Glover, life table, RNA sequencing, reproduction

## Abstract

The genus *Pseudomonas* encompasses a highly diverse array of bacteria that are ubiquitously present across diverse environments and hosts, with their effects ranging from beneficial to harmful. *Aphis gossypii* Glover shares a long-standing evolutionary connection with *Pseudomonas*, frequently establishing a symbiotic bond. *Pseudomonas* is intricately involved in numerous life processes of *A. gossypii* and significantly influences its physiological parameters. Despite the well-documented nature of this phenomenon, the underlying mechanisms remain obscure. In this study, to explore the relationship between *Pseudomonas* and *A. gossypii* in greater depth, we collected *A. gossypii* that were infected and non-infected with *Pseudomonas* from the wild. Subsequently, we analyzed the impact of *Pseudomonas* on *A. gossypii* by using life table parameters in combination with RNA-sequencing techniques. The results indicated that upon infection with *Pseudomonas*, the growth, development, and reproductive capabilities of *A. gossypii* were significantly and adversely affected.

## 1. Introduction

*Aphis gossypii* Glover, also known as the cotton aphid, is a globally distributed, polymorphic pest that feeds on over 300 species of host plants, particularly those in the *Cucurbitaceae*, *Solanaceae*, and *Brassicaceae* families [1]. The insect primarily damages crop yields through indirect feeding and the transference of viral diseases [2,3]. In recent decades, *A. gossypii* control has largely relied upon chemical insecticides, but efficacy is continuously threatened by the insect’s large populations, rapid reproduction, and high adaptability. Insecticide resistance in aphid populations negatively impacts the quality and safety of cotton production, resulting in serious chemical insecticide losses [4,5]. Consequently, many researchers and growers are turning to biological controls as a more environmentally friendly and reliable method of regulating *A. gossypii* [6,7].

Insects of herbivorous nature, specifically those possessing piercing-sucking mouthparts, subsist by consuming plant sap. Nevertheless, the nutrient content in the sap may not consistently fulfill their specific growth and developmental demands. Earlier scientific investigations have shown that endophytic bacteria have the potential to address this nutritional disparity [8]. *Buchnera*, the primary symbiotic bacteria in aphids, can furnish their hosts with a vast array of vital nutrients. For instance, in *Myzus Persicae* (green peach aphid), *Buchnera Mp* fixes atmospheric nitrogen and synthesizes essential amino acids such as leucine, isoleucine, and methionine [9,10]. These symbiotic bacteria fulfill indispensable functions in the survival, growth, reproduction of insects as well as their adaptation to host plants, natural predators, and high-temperature environments [11]. Oliver, K.M. et al. demonstrated that facultative symbiotic bacterial infection in *Acyrthosiphon pisum* (pea aphid) can significantly impede the growth and advancement of its natural adversary, *Aphidius service* Haliday [12]. Moreover, Attia, S. et al. observed that infection from the secondary symbiotic bacterium *Serratia* could diminish the pupation rate of *A. pisum* [13]. *A. pisum* containing *Serratia* also exhibits a higher reproduction rate and better suitability under high temperatures and heat stress conditions [14,15]. Another study demonstrated that *Drosophila* infected with *Wolbachia* experienced a significantly higher survival rate [16], along with a low probability of passing the infection on to their offspring [17]. Additionally, aphids infected with the parthenogenic symbiotic bacterium *Rickettsiella* have an increased content of blue-green polycyclic aromatic quinone, changing their body color from red to green [18]. Although secondary symbiotic bacteria are not essential for insect growth and reproduction, a growing body of research has demonstrated their role in bolstering insect defenses [19].

Nevertheless, the effects of symbiotic bacteria on insects are not invariably advantageous. For instance, *Pseudomonas*, a prevalent symbiotic bacterium, exerts notably negative effects on its hosts [20]. A 2016 study by DuPont Pioneer USA reported that the IPD072Aa protein produced by *Pseudomonas chlororaphis* exhibits insecticidal abilities against the western maize rootworm. Transgenic maize engineered with this protein is protected from the worm and contains important agronomic traits and nutrient composition [21]. Furthermore, the Fit protein produced by *Pseudomonas fluorescens* is virulent against *Manduca sexta* (tobacco hornworm) and *Galleria mellonella* (greater wax moth) [22]. Some strains of *P. protegens* and *P. chlororaphis* can infect and kill insect larvae after oral ingestion, potentially enabling rhizobia to resist grazing predators, thereby protecting the host and providing a competitive edge [23]. The bacterial genus can impact a wide variety of insect life activities. For example, *Pseudomonas* sp. form tiny, widespread, and stable communities in the gut of *Ceratitis capitata* including arthro-pathogenic bacteria. Injecting large amounts of *Pseudomonas* shortens the life span of *Ceratitis capitata*, while Enterobacteriaceae injections suppress *Pseudomonas* and extend the host’s life span [21,24]. Despite the growing body of research on symbiotic bacteria, the interactions between *Pseudomonas* and *A. gossypii* remain largely unexplored.

Recent research has facilitated the exploration of numerous biological functions of *Pseudomonas*, with a particular emphasis on its applications in biological control and plant growth promotion within the agricultural domain. In this study, we examined the growth and development indicators of *A. gossypii* infected with *Pseudomonas*, and then compared these with the pre-established life table parameters.

To gain a deeper understanding of the regulatory mechanisms of this bacterium on the life processes of *A. gossypii*, we employed RNA-Seq analysis to compare the gene expression dynamics between the infected and uninfected populations.

Overall, our findings shed light on the mechanisms underlying the use of *Pseudomonas* as a biological control agent against *A. gossypii*

## 2. Materials and Methods

### 2.1. Insects

*Aphis gopssypii* populations were originally collected from a cotton field at the Institute of Cotton Research, Chinese Academy of Agricultural Sciences (CAAS, 36°5′34.8″ N, 114°31′47.19″ E). Due to the fact that aphid symbiotes mainly propagate vertically from the mother to offspring during reproduction, they have a certain transmission failure rate [25]. Therefore, *A. gossypii* within five generations were used for data collection. *Pseudomonas* infection was determined according to the methods of [26]*. Pseudomonas* specific primers were designed based on the 16 s rRNA gene. qPCR was used to quantify copies of the *Pseudomonas* 16 s rRNA gene, with genomic DNA from a single aphid serving as the template. Detected gene copies were used to estimate the abundance of *Pseudomonas* in 1 μL of DNA solution. Aphid copy numbers exceeding the lowest point of the standard curve (~100 copies) were considered *Pseudomonas*-infected, while those below this threshold were considered uninfected.

### 2.2. Analysis of Biological Parameters

Treatments containing either 15 *Pseudomonas*-infected (there was no infection of other facultative symbiotic bacteria, or there were merely extremely low levels of these bacteria present) or non-infected (the organisms were either not infected with any facultative symbiotic bacteria or had very low levels of such bacteria). *A. gossypii* were reared in Petri dishes, and fed with the leaves of the cotton variety “Zhong 49”, with the non-infected *A. gossypii* serving as controls. These were maintained at 25 ± 1 °C, 65 ± 5% relative humidity (RH), and a 14:10 h light:dark photoperiod. The body length and width were recorded daily until adulthood. Individual body weight measurements began on the fourth day after emergence, and the experimental data were analyzed to detect differences among groups using a t-test, with GraphPad Prime 10.0 software serving as the analytical tool.

Treatments contained either 45 *Pseudomonas*-infected. The life table parameters of each offspring of *A. gossypii* were recorded individually. After recording, all of the offspring were removed, and the mother continued to give birth until the mother died. The life table data were first recorded in a TXT file. Subsequently, this file was uploaded to the relevant software, where the data were analyzed by means of the TWOSEX-MSChart program [27,28]. To obtain the means and standard errors, the bootstrap program was utilized, with 100,000 random resamples being carried out. To determine whether there were differences in the life table parameters between the treatment groups, paired bootstrap tests were employed [29].

### 2.3. RNA Extraction and RNA-Seq Analysis

Total RNA of adult *A. gossypii* was extracted with TRIzol (Thermo Fisher, Waltham, MA, 15596018CN) following the manufacturer’s instructions. The RNA quantity and quality were measured with a NanoDrop2000C spectrophotometer (Thermo Scientific, USA), and integrity was assessed by agarose gel electrophoresis.

cDNA libraries were constructed by sequencing extracted RNA at Shanghai Majorbio Bio-Pharm Technology Co. Ltd. (Shanghai, China) using the Illumina HiSeq X ten(NovaSeq X 1.3) sequencing platform. Clean reads were obtained by removing adaptors and low-quality reads. All raw data were uploaded in FASTQ format and stored in the NCBI SRA database under the accession number PRJNA832535. Gene expression levels were calculated as TPM values and differentially expressed genes (DEGs) were identified using the DESeq2 1.42.0 [30]. The isolated genes underwent assessment using DESeq2 1.42.0 software, employing a stringent threshold of *p* < 0.05 with |log2(fold change)| > 2 criteria for characterizing DEGs. Relationships between samples were assessed through principal component analysis. Functional enrichment analyses including Gene Ontology (GO) and Kyoto Encyclopedia of Genes and Genomes (KEGG) were performed using Goatools and KOBAS [31] to analyze the DEG functions. GO and KEGG terms were considered significantly enriched at a corrected *p*-value < 0.05.

### 2.4. RT-qPCR Analysis

DEGs identified from the RNA-Seq data were analyzed by RT-qPCR. A template consisting of 1 μg of RNA was utilized for reverse transcription using the PrimeScript™ RT Reagent Kit with gDNA Eraser (TAKARA Bio, Beijing, China, RR047Q) according to the manufacturer’s instructions. The RT-qPCR mixture comprised 5 μL Trans-Start Top Green qPCR SuperMix (Dye I) (2×, TransGen, Beijing, China, AQ131-01), 0.2 μL of each primer (10 μM), 1 μL of cDNA (20× dilution), and RNase-free water up to 10 μL. Real-time PCR reactions were performed on a Step OnePlus™ Real-Time PCR System (Applied Biosystems, Foster City, CA, USA) using the two-step method: initiation at 95 °C for 30 s, followed by 40 cycles of 95 °C for 5 s and 60 °C for 30 s. Three biological and three technical replicates were performed for each RT-qPCR reaction. Relative expression levels of the target genes were calculated using the 2^−ΔΔCt^ method [32]. Elongation factor1 alpha (*EF1α*) and beta-actin (*β-ACT*) genes were used as the endogenous reference gene for normalizing the target gene expression levels [33]. All primers used in this study are listed in Table 1.

## 3. Results

### 3.1. Aphid Body Length and Width

The body lengths of *A. gossypii* both infected and uninfected with *Pseudomonas* (*Pse* and no-*Pse*) exhibited an increasing trend, plateauing on the third day of the adult (Figure 1A). The *Pse* group was marginally shorter than the control, but this difference was not statistically significant. Neither population showed significant increases in length following the adult 3 d timepoint, however, the no-*Pse* was markedly longer than *Pse*. The body width of *A. gossypii* in both populations demonstrated an ascending trend prior to the fifth day of adulthood. From adult 5 to 15 days, the growth of the body width of *A. gossypii* in both populations tended to plateau, and there was no significant disparity in body width between *Pse* and no-*Pse* (Figure 1B).

### 3.2. Aphid Weight

Body mass measurements of single *A. gossypii* were recorded daily, starting at the fourth instar. The body weights from the first to third instars were too low to measure with the balance. After the 11th day, weights were recorded every other day (Figure 1C).

From 4L (fourth instar) to 5 d, the weight of the aphids in both treatment groups consistently increased, reaching peak values at 5 d (no-*Pse* = 24.13 ± 5.12, *Pse* = 19.47 ± 3.23) (Figure 1C). With the exception of 4L, the *Pse* group weighted significantly less than the no-*Pse* control at all stages.

### 3.3. Aphid Offspring Size

Both *A. gossypii* treatment populations began reproducing on the first day of adulthood, with birth numbers progressively increasing between 1 and 4 d and peaking from 4–7 d. During this time, the daily mean birth number surpassed four. After the 7th day, the birth number steadily diminished, with no additional offspring produced after 20 days (Figure 1D). The total number of *Pse* offspring (36.75 ± 9.79) was significantly lower than the no-*Pse* treatment (44.7 ± 10.91), with a difference of 7.95 (t = 2.710, df = 49, *p* = 0.0092) (Figure 1E).

### 3.4. Aphid Life Table Parameters

We assessed the life tables of the *A. gossypii* populations infected and uninfected with *Pseudomonas* (Table 2). The *Pse* group had a finite increase rate of 1.52 and an innate increase rate of 0.42, both showing no significant difference compared with the no-*Pse* group, respectively. However, the longevity of adult and lifespan, fecundity, net reproductive rate *R*_0_, and mean generation time *T*(d) were significantly lower than that of the no-*Pse*. This indicates that *Pseudomonas* infection negatively impacts *A. gossypii* reproduction and lifespan.

### 3.5. Transcriptomic Analysis

A total of 259,170,048 raw sequences were generated by Illumina Hiseq technology, resulting in 254,733,654 pristine sequences following stringent quality control and filtering. Q20 and Q30 quality scores exceeded 98.48% and 95.25%, respectively. The reference genome alignment rate was above 54.03% (Table 3). The GC rate ranged from 34.5% to 43.14%. These results underscore the superior quality of the sequencing data, reinforcing the reliability of subsequent gene expression analyses. The expressed genes were annotated with the GO, KEGG, COG, NR, Swiss Prot, and Pfam databases, resulting in classifications of 6249, 7119, 11,523, 12,621, 9192, and 10,504 genes, respectively (Figure 2A).

### 3.6. DEG Functional Enrichment Analysis

The principal component analysis (PCA) graphs of the transcriptomic sequencing data illustrated considerable differences between *Pse* and no-*pse A. gossypii* populations, with compact clustering within each treatment group (Figure 2B). A volcano plot was used to depict the contrast in gene expression levels between treatments, identifying 255 DEGs across all samples (*p* < 0.05). Compared with the control, the *Pse* group exhibited 181 upregulated and 74 downregulated DEGs (Figure 2C). These results indicate that significant transcriptional differences in *A. gossypii* are induced by *Pseudomonas* infection.

A GO enrichment analysis was conducted on significant DEGs to classify their functions, resulting in three major domains: biological process (BP), cellular component (CC), and molecular function (MF). The MF classification primarily includes monooxygenase activity, iron ion binding, and serine-type peptidase activity. BP encompasses processes related to carbohydrate metabolism, oxidation–reduction, and cellular amino acid biosynthesis. CC primarily consists of membrane components (Figure 3A).

We examined the metabolic pathways involved in the DEGs using the KEGG database. A total of 173 DEGs showed significant changes between the two cotton aphid populations, mainly concentrated in 12 metabolic pathways (*p* < 0.05). The most important pathways were related to carbohydrate digestion and absorption, galactose metabolism, valine, leucine and isoleucine biosynthesis, and lysosome processes (Figure 3B).

### 3.7. Validation of RNA-Seq Results

To validate the quality and accuracy of the transcriptome data, eight differentially expressed genes (DEGs) were randomly selected for reverse transcription-quantitative polymerase chain reaction (RT-qPCR). The outcomes of this analysis demonstrated the upregulation of genes encoding endocuticle structural glycoprotein SgAbd-2, UDP-glucuronosyltransferase, 60S ribosomal protein L11, 40S ribosomal protein S17, hemolymph juvenile hormone-binding protein (JHBP), and cytochrome P450 4c1. In contrast, the fatty acid synthase and vitellogenin genes were downregulated. These observations were consistent with the results from RNA-sequencing (RNA-Seq), as illustrated in Figure 4.

In the transcriptome analysis, fragments per kilobase of exon model per million mapped read (FPKM) values were utilized to quantify the gene expression levels. The results of the quantitative validation were in line with the trends in the gene expression levels observed in the transcriptome, thus indicating the reliability of the transcriptome results.

## 4. Discussion

The diverse and widespread presence of microorganisms has led to the development of complex relationships with insects throughout evolution. The metabolism and physiological activities of insects are influenced by microorganisms in many ways, both directly and indirectly [34,35]. For instance, monophagous aphids that feed on plant sap often rely on symbiotic bacteria to digest plant polysaccharides and provide sources of nutrients such as carbon, essential amino acids, and vitamins [36]. This mutualistic relationship is typically observed in nutrient-poor environments [37]. Previous studies have also suggested that certain symbiotic bacteria may adversely affect the growth and advancement of their hosts. Garcia, L.C. et al. [38] determined that *heterohadididae* and *Steinernematidae* nematodes dispense *Photorhabdus* and *Xenorhabdus* bacteria to annihilate the host insect, the fall armyworm (*Spodoptera frugiperda*). Moreover, Thanwisai A et al. [39] reported that these nematode families infected with *Photorhabdus* can lethally impact *Aedes aegypti* (yellow fever mosquito) and *Culex quinquefasciatus* (southern house mosquito). The beneficial bacteria *Pseudomonas* is commonly found in human intestines; however, increasing research has shown its potential to harm insect hosts [20,40]. Vodovar N et al. reported that *Pseudomonas* infection causes damage to the intestinal cells of *Drosophila* larvae [41]. Our understanding of aphid–microbe relationships has advanced significantly, with particular emphasis on the roles of *Pseudomonas* in biological control. Previous studies have also found that *Pseudomonas* has adverse effects on insect hosts. *Pseudomonas*, which colonizes plants, can invade insects when orally ingested, leading to the death of susceptible pest insects [42]. The bacterium *Pseudomonas* achieved a kill rate of over 90% of *Culex pipiens* and *Aedes albopictus* larvae within 72 h after exposition to a bacterial concentration of 100 million CFU/mL [43]. However, their roles in the life processes of cotton aphids remain unclear. This study analyzed the impact of *Pseudomonas* infection on *A. gossypii* using life table parameters and transcriptome sequencing, advancing our understanding of their symbiosis.

Numerous reports have elucidated how symbiotic bacteria affect various insect traits related to reproduction, growth, and development [44]. Our results indicate that *Pseudomonas* infection in *A. gossypii* significantly suppresses the host’s reproductive capacity and offspring size. Additionally, the rates of endogenous and circumferential growth were significantly reduced in the *Pseudomonas*-infected aphids. *Pseudomonas* also disrupted the normal growth and development of *A. gossypii*, causing a decrease in aphid length, width, and body weight.

Symbiotic bacteria are known to greatly affect insect reproduction, influencing their hosts in various ways that may be beneficial or harmful [45]. For example, *Rickettsia* bacteria play a regulatory role in *Bemisia tabaci* (silverleaf white fly) reproduction. High proportions of infected females exhibit increased offspring sizes and higher overall survival rates [46]. Additionally, *Wolbachia* bacteria remarkably manipulate insect reproduction, greatly increasing the female-to-male ratios in populations of *Culex pipiens* (common mosquito) [47]. In *Prostephanus truncatus* (larger grain borer), *Wolbachia* leads to reduced offspring sizes and cytoplasmic incompatibility [48]. Finally, aphids infected with *Serratia* tend to exhibit reduced viability and fertility [49].

Likewise, symbiotic bacterial infections can either positively or negatively affect the growth and development of insect hosts. Treating *Eurygaster integriceps* (Sunn pest) with the antibiotic norfloxacin has been shown to significantly impair the growth and development of offspring, indicating the important roles of symbiotic bacteria [50]. Conversely, *A. pisum* infected with *Hamiltonella* and *A. gossypii*s infected with *Serratia* experience weight loss, suggesting an adverse relationship [51,52]. These examples highlight the variable influences of symbiotic bacteria on the reproductive, growth, and developmental processes of insects. We hypothesize that the changes observed in our treatment groups could be attributable to the pathogenic nature of the microorganism [20]. Pathogenic bacteria typically enter insects through ingestion or injection into the epidermis, where they release toxins or other pathogenic factors [53]. The intricate interactions and the metabolites produced by *Pseudomonas* provide a promising pathway for biological control strategies.

In this study, we conducted an RNA-Seq analysis to compare the physiological and biochemical differences between *Pseudomonas*-infected and non-infected *A. gossypii*. Genes enriched in the KEGG pathway were concentrated in UGT, LCT, hexokinase, alkaline phosphatase, and cathepsin B. The upregulation of UGT was mainly through pentose and glucuronate interconversions as well as porphyrin and chlorophyll metabolism. UDP-glucuronosyl transferase enhances the activity of the insects’ intrinsic detoxification enzymes, diminishing their susceptibility to insecticides [54,55]. The upregulation of the lactase rhizosphere hydrolase (LCT) in the KEGG pathway is believed to modify the plasticity of the insect’s epidermis and assist host adaptability. While elevated levels of LCT enhance the growth rate and overall performance of *Oedaleus asiaticus* (common grasshopper), they also lead to a reduction in insect body size [56]. Our enrichment analysis revealed that an alkaline phosphatase gene was downregulated in the folate synthesis pathway under *Pse* treatment. Alkaline phosphatase is a ubiquitous enzyme with distinct roles across various tissues and organs. This enzyme is implicated in insect nutrient uptake, midgut lumen alkalinization, development, neurological and renal function, and cuticle sclerosis [57]. In aphids, alkaline phosphatase influences detoxification and host defense/manipulation. Further research is required to clarify the roles of alkaline phosphatase in the saliva of hemipteran insects [58]. In our study, cathepsin B was downregulated in the NOD-like receptor signaling pathway. Cathepsin B is an intracellular protease predominantly located in lysosomes, whose presence is indispensable for the vital activities of the organism. During embryonic development, the enzyme degrades yolk proteins to provide essential amino acids [59]. Nutrient catabolism is a crucial biochemical process during the early stages of insect embryonic development, with the inhibition of protease activity during this period leading to early egg termination. Our results suggest that the initial size and weight decrease observed in *A. gossypii* infected *Pse* may be attributed to the downregulation of *UGT* and *LCT*. Conversely, the downregulation of alkaline phosphatase and cathepsin B may explain the significant reduction in offspring size, endogenous growth rate, and perinatal growth rate after *Pseudomonas* infection.

## 5. Conclusions

This study presented a comprehensive analysis of the effects of *Pseudomonas* on *A. gossypii.* Our results demonstrate that *Pseudomonas* infection disrupts the growth and development of *A. gossypii*, significantly reducing the offspring size, innate rate of increase, and finite rate of increase. Additionally, the bacteria affects the expression of genes such as UGT and LCT, which are primarily involved in pentose and glucuronate interconversions, porphyrin and chlorophyll metabolism, galactose metabolism, and carbohydrate digestion and absorption pathways. This work provides a theoretical basis for microbial pest control strategies and furthers our understanding of insect–microbe interactions.

## Figures and Tables

**Figure 1 insects-16-00238-f001:**
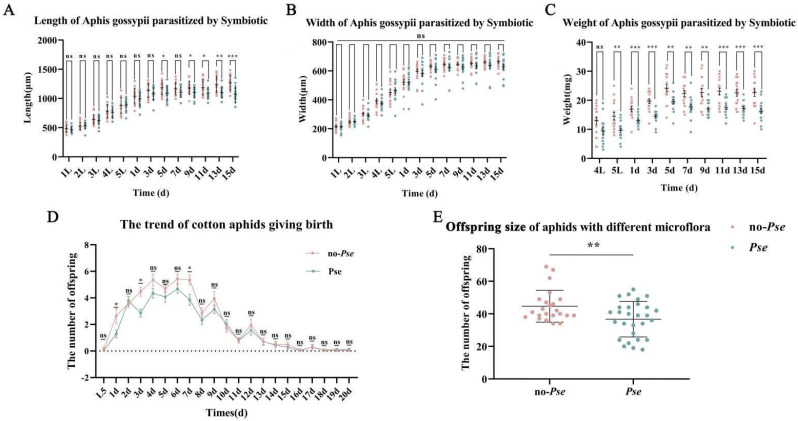
Influence of *Pseudomonas* on *A. gossypii* physical characteristics. Insect body (**A**) length, (**B**) width, (**C**) weight, (**D**) trend chart of daily litter size of *A. gossypii*, and (**E**) offspring size. Data represent the mean and standard error. SAS V8 software was used to analyze the body length and width through a one−way ANOVA. *: *p*-value ≤ 0.05, **: *p*-value ≤ 0.01, ***: *p*-value ≤ 0.001, and ns indicates no significant difference.

**Figure 2 insects-16-00238-f002:**
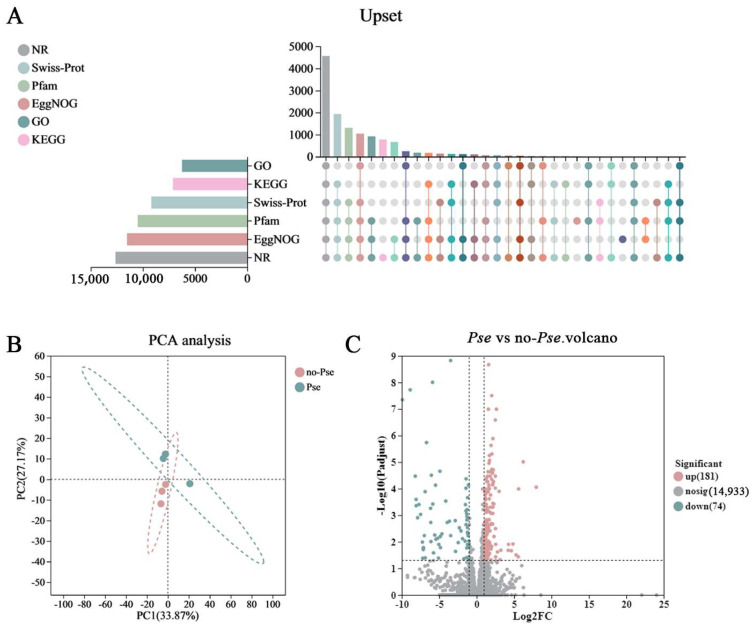
(**A**) Expression gene annotation map of *Pseudomonas*. (**B**) Principal component analysis of the transcriptome samples. (**C**) Volcano plot of DEGs between the infected and uninfected aphid populations.

**Figure 3 insects-16-00238-f003:**
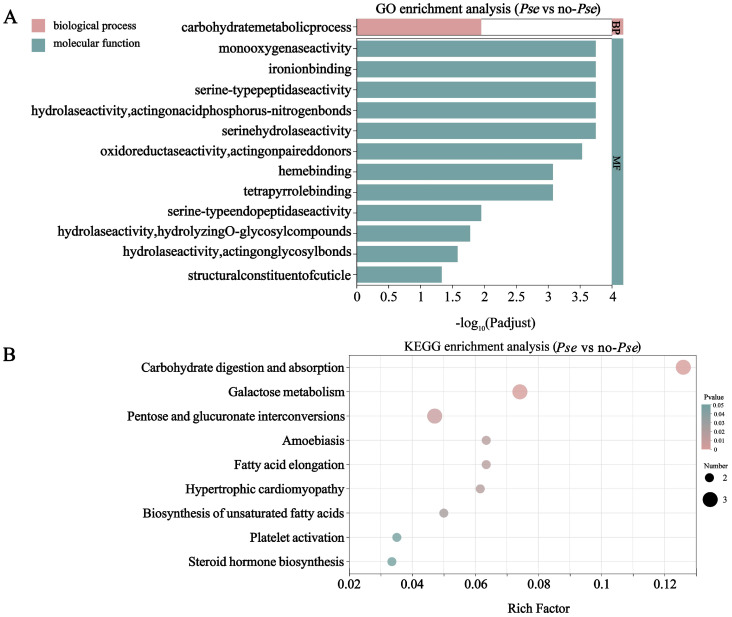
GO and KEGG pathway enrichment analysis of DEGs. (**A**) Bar chart of GO annotation for enriched pathways of differentially expressed genes between the *Pse* and the no-*Pse*. (**B**) Bubble chart of KEGG pathway annotations for the enriched differentially expressed genes between the *Pse* and the no-*Pse*.

**Figure 4 insects-16-00238-f004:**
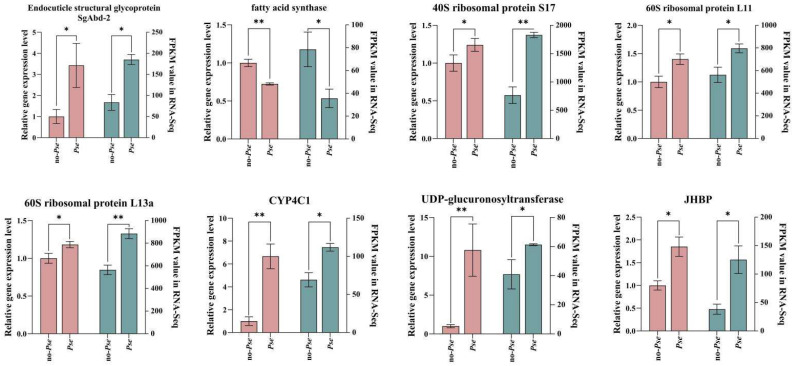
qRT-PCR validation of 8 selected DEGs. For each bar chart, the left-hand side shows the detection result of RT-qPCR, while the right-hand side presents the FPKM value from the transcriptome results. These values serve to reflect the gene expression levels. Columns and bars represent the means and standard errors, respectively. Stars indicate statistical significance (*: *p*-value ≤ 0.05, **: *p*-value ≤ 0.01).

**Table 1 insects-16-00238-t001:** Encoding genes and corresponding sequences of the qRT-PCR primers.

Prime Name	Gene Name	Primer Sequences (5′–3′)
EF1a-F	Actin	GAAGCCTGGTATGGTTGTCGT
EF1a-R		GGGTGGGTTGTTCTTTGTG
LCT-F	Lactase rhizosphere hydrolase	GCTGATGTGTATAAGGGCATGGGAG
LCT-R		AATCCGCAGCAATATCTCCGTTGAA
UGT-F	UDP-glucuronosyl transferase	TCCCTTCGCCAATGGTCTCCAA
UGT-R		TGTTCTAGGCACCGCGTGATGA
Vg-F	Vitellogenin	GCACGAGCCATAATTGTTGAG
Vg-R		ACCCGGTTTCATGGTTGGT
SgAbd-2-F	Endocuticle structural glycoprotein SgAbd-2	ACGTCGTACGTCACGAATACA
SgAbd-2-R		GGAGACCGCGAGGAAAGAAA
JH-F	Hormone binding protein (JHBP)	AGCTGCACTTATTATCTATAGTTGT
JH-R		ATTGTCCGTTCCGTCAATCG
FAS-F	Fatty acid synthase	TCGCGATCATTGTTATGGTCCT
FAS-R		TCAAACCACATTTGTCTGAACAGT
60S-F	60S ribosomal protein L13a	CTGGTAACTTCGGTTTCGGT
60S-R		AAATAAGATAAGCTAACCTTGACG
40S-F	40S ribosomal protein S17	CTGCCAGAGTCATCATCGAG
40S-R		TCATATTTAGTATTTACCCAGCGA
CYP4c1-F	Cytochrome P450 4c1	TAAAACAACTTCAGGGGTGG
CYP4c1-R		ACAATGATGGTAAGTTTTTGAGTT

Primer name refers to the name assigned to the designed primer; Gene name represents the name of both the reference gene and the gene employed for quantitative validation; On the far-right side of the table are the upstream and downstream primers of the corresponding gene.

**Table 2 insects-16-00238-t002:** The effect of Pseudomonas on the life table parameters of *A. gossypii*.

Parameters	no-*Pse* (*n* = 45)	*Pse* (*n* = 45)	Significant Level	*p*-Value
Adult longevity (d)	20.72 ± 0.51	15.54 ± 0.80	****	<0.00001
Total longevity (d)	24.72 ± 0.0.51	19.54 ± 0.0.80	****	<0.00001
Fecundity (nymphs)	45.50 ± 2.37	36.71 ± 2.02	**	0.00449
Oviposition days (d)	11.72 ± 0.66	10.79 ± 0.52	ns	0.26517
*R*_0_ (offspring)	45.49 ± 2.37	36.71 ± 2.02	**	0.00449
*r* (d^−1^)	0.40 ± 0.01	0.42 ± 0.01	ns	0.20238
*λ* (d^−1^)	1.49 ± 0.02	40.50 ± 2.08	ns	0.19647
*T* (d)	9.53 ± 0.02	8.60 ± 0.17	***	0.0004

Values in the table represent the mean ± SE. The data at the significant level represent the level of significant difference between Group A and Group B, where the meaning is: ns: *p*-value > 0.05, **: *p*-value ≤ 0.01, ***: *p*-value ≤ 0.001, ****: *p*-value ≤ 0.0001. Adult longevity (d) is the lifespan of A. gossypii after emergence; total longevity (d) is the entire life cycle of A. gossypii; fecundity is the offspring number per female; oviposition days (d), the interval between the first and last oviposition of A. gossypii.; *R*_0_, net reproductive rate (offspring per individual); *r*, intrinsic rate of increase (d^−1^); *λ*, finite rate of increase (d^−1^); *T*, mean generation time (days).

**Table 3 insects-16-00238-t003:** Quality control data table of *A. gossypii* transcriptome.

Sample	Raw Reads	Clean Reads	Error Rate (%)	Q20 (%)	Q30 (%)	GC Content (%)	Mapped
no-*Pse* 1	44,619,326	43,910,728	0.0235	98.68	95.69	36.75	77.45%
no-*Pse* 2	43,081,224	42,564,944	0.0234	98.71	95.72	36.91	54.03%
no-*Pse* 3	42,073,736	41,477,186	0.0229	98.88	96.27	43.14	70.42%
*Pse* 1	42,825,858	41,983,916	0.0234	98.71	95.78	36.51	90.61%
*Pse* 2	41,839,382	40,938,182	0.0234	98.68	95.75	36.21	87.82%
*Pse* 3	44,730,522	43,858,698	0.0239	98.48	95.25	34.5	81.29%

## Data Availability

All data contained within the article.

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
