# Peer review of "Pseudomonas Infection Affects the Growth and Development of Aphis gossypii by Disrupting Energy Metabolism and Reproductive Processes"

_insects, 2025, doi:10.3390/insects16030238_

Round 1
Reviewer 1 Report
Comments and Suggestions for Authors
Comments to Authors:
This study deals with the effects of Pseudomonas infection on growth and development of Aphis gossypii, and further impacts on transcription. Authors stated a novel interaction between aphid and symbiotic bacteria. Their results will be helpful to utilize symbiotic bacteria in biological control of aphids. However, the manuscript was written badly, mainly reflect in citation formats, omission of important tables. Many mistakes can be found in this manuscript, I strongly doubt whether the authors wrote and reviewed the manuscript carefully before submitting it. Particularly, just 15 individuals were used to construct a life table. For constructing a life table of insects, at least 30 aphids are needed. Moreover, genes with an “adjusted P-value < 0.05” estimated by DESeq were assigned as differentially expressed, not “P-value < 0.05”. Based on these issues, the results throughout manuscript are suspect. Therefore, I think that this manuscript has not met the requirements for acceptance. Some important comments could be seen below.
The tables in ms are confusing:
Line 156-157: “All primers used in this study are listed in Table 1.”. But, Table 1 showed the effect of Pseudomonas on A. gossypii life table parameters, please see Line 159. Therefore, I can not find the primer sequences used in this study.
Line 189: the Table 2 can’t be found in ms.
Line 198: the Table 3 can’t be found in ms
Line 60: “A. service”
Line 102: Please provide the geographical coordinates. Another, if these aphid populations were reared in laboratory? If so, how many generations were successively reared?
Line 113: I think that 15 individuals were not enough to establish a life table.
Line 119-120: What data was analyzed using a one-way ANOVA with SAS software?
Line 121: Utilize the latest version of the TWOSEX program.
Line 122-123: Authors are encouraged to cite Efron and Tibshirani (1993).
Line 124: What software was utilized to analyze the differences in life table parameters?
Line 126: “Total RNA of adult aphid”.
Line 136: “P-values <0.05” was directly used to selected significant differential expression? If so, the data will not be reliable.
Line 153: “(K.J. Livak and T.D. Schmittgen, 2001)” should not be superscript.
Line 155-156: Differences in relative gene expression were analyzed by comparing “ΔCt values”? So why 2-ΔΔCt method was used to calculate relative expression levels of the target genes?
Line 159: Many problems can be observed in Table 1. (1) The number of aphids used in this study should be clarified in Table. (2) The first parameter is confusing, adult? Adult longevity? But the second parameter is longevity. (3) The representation of difference is also not quite appropriate, I think “*” is much better here. (4) The P value should be added in Table. (5) “R0”, Ensure that “R” is italicized, and "0" in R0 is not italicized. (6) The units should be added behind the first five parameters. (7) The nymph duration should also be added in Table 1. (8) A necessary note for Table 1 is lacking.
Line 172-179: How many aphids in each group were used in body mass measurements? Also, the weight here is “individual weight” or “weight of 100 aphids” or others? Please clarify in this section and Materials and Methods.
Line 189: The Pse group had a finite increase rate of 1.52? I think that the data in Table, 40.5, is wrong.
Line 189-192: Based on these sentences, I strongly doubt whether the authors reviewed the manuscript carefully before submitting it. Authors wrote out the values of finite increase rate and innate increase rate, so the units is necessary. Also, r and λ should be added behind innate increase rate and finite increase rate, respectively. The initial letter of these three parameters, Fecundity, Net reproductive rate R0, and Mean generation time T(d), should be lowercase. The “(d)” behind mean generation time T should be deleted, because authors did not write the value. The P value is missing through this paragraph.
Line 198: Where is Table 3?
Line 210: “A. gossypii”, italic.
Line 224: “randomly”
Line 222-228: In Fig. 4, authors just exhibited the results of RT-qPCR, they did not show the RNA-seq results. So, Fig. 4 should be re-construction, this paragraph should be rewritten as well.
Line 250: “Pseudomonas”, italic.
Line 263-264: Was the decrease of length, width, and body weight associated with the disruption of the normal growth and development of aphids?
Line 279: Here, Acyrthosiphon pisum is not the first time to state, so it should be written as “A. pisum”.
Line 370: “2009, Leaf-nosed bat, Journal, Issue,”, confusing.
Line 369: I have not seen the references formats like this. Authors are encouraged to revise the formats according to “Instructions for Authors”.
Author Response
Comments and Suggestions for Authors
Comments to Authors:
This study deals with the effects of Pseudomonas infection on growth and development of Aphis gossypii, and further impacts on transcription. Authors stated a novel interaction between aphid and symbiotic bacteria. Their results will be helpful to utilize symbiotic bacteria in biological control of aphids. However, the manuscript was written badly, mainly reflect in citation formats, omission of important tables. Many mistakes can be found in this manuscript, I strongly doubt whether the authors wrote and reviewed the manuscript carefully before submitting it. Particularly, just 15 individuals were used to construct a life table. For constructing a life table of insects, at least 30 aphids are needed. Moreover, genes with an “adjusted P-value < 0.05” estimated by DESeq were assigned as differentially expressed, not “P-value < 0.05”. Based on these issues, the results throughout manuscript are suspect. Therefore, I think that this manuscript has not met the requirements for acceptance. Some important Comments could be seen below.
Dear reviewer:
Thank you for reviewing the manuscript and providing constructive feedback.
We sincerely apologize for the errors and problems in the manuscript that have adversely affected the reviewers' ability to read and understand our experimental results. We are also extremely grateful to the reviewer for their efforts in identifying these issues. In response to the two major concerns raised by the reviewer, we would like to offer the following explanations.
First, regarding the sample - size issue in the life - table analysis. We initiated the experiment with 60 aphids and proceeded with life - table analysis. During this process, samples that died abnormally were excluded. Eventually, 45 cotton - aphid samples were utilized for the life - table data analysis. It should be noted that the "15 Pseudomonas - infected" mentioned in the manuscript were solely employed for the sample - size analysis of body - length, width, and weight data.
Second, concerning the criteria for screening differentially expressed genes. In fact, the criterion we adopted is 'The isolated genes underwent assessment using DESeq software, employing a stringent threshold of p < 0.05 with |log2(fold change)| >2 criteria for characterizing DEGs.' for the screening of differentially expressed genes.
These two key issues originated from writing - related mistakes rather than problems in the experimental procedures. We have made comprehensive revisions to the manuscript. Additionally, we have thoroughly re - examined the manuscript content, identified, and rectified all the errors present.
Once again, we are truly sorry for any inconvenience our negligence may have caused to your understanding.
Comments 1: Line 156-157: “All primers used in this study are listed in Table 1.”. But, Table 1 showed the effect of Pseudomonas on A. gossypii life table parameters, please see Line 159. Therefore, I can not find the primer sequences used in this study.
Response 1: Thank you for your comments. We agreewith this comment. This was an error in table insertion. We have thoroughly reviewed and re - sequenced the tables in the manuscript. Specifically, Table 1 (the Primer Table) has been inserted at the 184th line of the manual correction.
Comments 2: Line 189: the Table 2 can’t be found in ms.
Response 2: Thank you for your comments. We agreewith this comment. We have inserted Table 2 into line 226 of the manuscript.
Comments 3: Line 198: the Table 3 can’t be found in ms
Response 3: Thank you for your comments. We agreewith this comment. We have inserted Table 3 into line 241 of the manuscript.
Comments 4: Line 60: “A. service”
Response 4: Thank you for your comments. We agreewith this comment. Therefore, according to the revision suggestions, we have changed “Aphidius service Haliday” to “A. service” on Line 74 of manuscript.
Comments 5: Line 102: Please provide the geographical coordinates. Another, if these aphid populations were reared in laboratory? If so, how many generations were successively reared?
Response 5: â‘ Thank you for your comments. We agreewith this comment. Therefore, according to the revision suggestions, The specific coordinate information of the experimental site has been added after the abbreviation on Line 118 of manuscript;
â‘¡ In the laboratory, cotton aphids reared over multiple generations experience a gradual decline in symbiotic bacteria. Moreover, symbiotic bacteria in different cotton aphid populations show signs of convergence. To address this, we sourced wild cotton aphid populations. After bringing them into the laboratory, we expanded their numbers before commencing experiments. It should be noted that all the aphids utilized in the experiment were those that had been reared in the laboratory for fewer than five generations, Prior to and during the experiment, quantitative verification of the symbiotic bacteria in cotton aphids will be carried out to ensure the accuracy of the experiment. And the relevant content has been incorporated into the "Materials and Methods" section of the manuscript.
Comments 6: Line 113: I think that 15 individuals were not enough to establish a life table.
Response 6: In this paragraph, we failed to clearly describe the specific procedures of the research method employed. Given our concern that excessive handling or movement of cotton aphids could have negative impacts on their growth and development, we carried out two separate sets of experiments. One set was dedicated to collecting data on body weight, width, and length, while the other was for gathering life - table data.
For the experiment on body weight, width, and length, we initially selected 20 cotton aphids for data analysis. However, we excluded samples that died abnormally. Eventually, 15 cotton aphids were chosen as the final research samples for this part of the study.
Regarding the life - table data collection, we selected 60 cotton aphids as the research subjects. After eliminating the samples with abnormal deaths, 45 cotton aphids were ultimately determined as the final research samples.
In the manuscript, the term '15 Pseudomonas - infected' specifically refers to the cotton aphids utilized for collecting data on body length, width, and weight. Due to an oversight on our part, we neglected to describe the cotton aphids involved in collecting life - table data. We sincerely apologize to the reviewer for the confusion and misunderstanding this error has caused. The manuscript has been revised to present the correct research methods accurately.
Comments 7: Line 119-120: What data was analyzed using a one-way ANOVA with SAS software?
Response 7: Thank you for pointing this out. We used one-way ANOVA in SAS to analyze the data of body length, body width, and body weight. However, after consultation and discussion, we decided to use T-test in Graphped Prime 10.0 software to analyze and plot these data. The reanalysis results showed no difference from the previous analysis results.
Comments 8: Line 121: Utilize the latest version of the TWOSEX program.
Response 8: Thank you for pointing this out. We agreewith this comment. In response to the suggestion, we will utilize the most recent version of the TWOSEX software for the analysis of life-table data. There exist certain minor discrepancies in the experimental data analyzed by the two versions of the software. However, these differences do not exert any influence on the analytical outcomes of the experiment.
Comments 9: Line 124: What software was utilized to analyze the differences in life table parameters?
Response9: Thank you for your comments. We have opted to re-utilize the latest version of the TWOSEX program for calculating life-table data. Moreover, the differences in life-table parameters were directly presented using the results computed by the program.
Comments 10: Line 122-123: Authors are encouraged to cite Efron and Tibshirani (1993).
Response 10: Thank you for your comments. We agreewith this comment. Therefore, we have cited the valuable reference on Line 149 of the manuscript.
Comments 11: Line 126: “Total RNA of adult aphid”.
Response 11: Thank you for your comments. We agreewith this comment. Therefore, we have change “Total adult aphid RNA” to “Total RNA of adult aphid” on Line 151.
Comments 12: Line 136: “P-values <0.05” was directly used to selected significant differential expression? If so, the data will not be reliable.
Response 12: Thank you for your comments. We agreewith this comment. In the manuscript, we were not accurately describe the screening criteria for differences. The isolated genes underwent assessment using DESeq software, employing a stringent threshold of p < 0.05 with |log2(fold change)| >2 criteria for characterizing DEGs. Therefore, we have made revisions and corrections to the manuscript no Line 161-163.
Comments 13: Line 153: “(K.J. Livak and T.D. Schmittgen, 2001)” should not be superscript.
Response 13: Thank you for your comments. We agreewith this comment. In the manuscript, the superscript for (K.J. Livak and T.D. Schmittgen, 2001) has been removed on Line 179.
Comments 14 Line 155-156: Differences in relative gene expression were analyzed by comparing “ΔCt values”? So why 2-ΔΔCt method was used to calculate relative expression levels of the target genes?
Response 14: Thank you for your comments. The error that occurred here can be attributed to the lack of clarity in our original expression. When it came to detecting differential gene expression levels, we procured the ΔCt value by means of RT-qPCR. Then, we determined the final outcome by applying the 2-ΔΔCt calculation approach. In light of this, we have made revisions to this paragraph on Line 178-179 to ensure that the expression was more precise.
Comments 15: Line 159: Many problems can be observed in Table 1. (1) The number of aphids used in this study should be clarified in Table. (2) The first parameter is confusing, adult? Adult longevity? But the second parameter is longevity. (3) The representation of difference is also not quite appropriate, I think “*” is much better here. (4) The P value should be added in Table. (5) “R0”, Ensure that “R” is italicized, and "0" in R0 is not italicized. (6) The units should be added behind the first five parameters. (7) The nymph duration should also be added in Table 1. (8) A necessary note for Table 1 is lacking.
Response 15: Thank you for your constructive feedback. In consideration of the above-mentioned seven questions, we have constructed a new life table. Moreover, we have made both modifications and supplements to the content of the previous one on Line 226.
Comments 16: Line 172-179: How many aphids in each group were used in body mass measurements? Also, the weight here is “individual weight” or “weight of 100 aphids” or others? Please clarify in this section and Materials and Methods.
Response 16: Thank you for your comments. Here, the term “weight” specifically denotes “individual weight.” This value is obtained by measuring the mass of a single cotton aphid with the use of a high - precision balance. We made revisions on line 137 and 200 of the manuscript.
Comments 17: Line 189: The Pse group had a finite increase rate of 1.52? I think that the data in Table, 40.5, is wrong.
Response 17: Thank you for your comments. Through meticulous comparison, we discovered that an error was made in filling in the table contents, and this has already been rectified in the re-edited life table on Line 226 of the manuscript.
Comments 18: Line 189-192: Based on these sentences, I strongly doubt whether the authors reviewed the manuscript carefully before submitting it. Authors wrote out the values of finite increase rate and innate increase rate, so the units is necessary. Also, r and λ should be added behind innate increase rate and finite increase rate, respectively. The initial letter of these three parameters, Fecundity, Net reproductive rate R0, and Mean generation time T(d), should be lowercase. The “(d)” behind mean generation time T should be deleted, because authors did not write the value. The P value is missing through this paragraph.
Response 18: Thank you for your comments. We wholeheartedly embrace your comments and suggestions. Owing to our oversight, we failed to meticulously review the errors present in the life chart. Consequently, we have re-created the life chart and rectified the emerging errors. Moreover, in light of your feedback, we have supplemented and revised the content of the life table.
Comments 19: Line 198: Where is Table 3?
Response 19: Thank you for your comments. We have inserted Table 3 at line 241 of the manuscript.
Comments 20: Line 210: “A. gossypii”, italic.
Response 20: Thank you for your comments. We agreewith this comment. The term “A. gossypii” has been italicized on line 251 of the manuscript.
Comments 21: Line 224: “randomly”
Response 21: Thank you for your comments. We agreewith this comment. We have made modifications to 'randomly' on line 267 of the manuscript.
Comments 22: Line 222-228: In Fig. 4, authors just exhibited the results of RT-qPCR, they did not show the RNA-seq results. So, Fig. 4 should be re-construction, this paragraph should be rewritten as well.
Response 22: Thank you for your comments. We agreewith this comment. We have re-constructed a validation figure for differential gene expression levels, leveraging the results from RT - qPCR and transcriptome sequencing.
Comments 23: Line 250: “Pseudomonas”, italic.
Response 23: Thank you for your comments. We agreewith this comment. We have made modifications to 'Pseudomonas' on line 301 of the manuscript.
Comments 24: Line 263-264: Was the decrease of length, width, and body weight associated with the disruption of the normal growth and development of aphids?
Response 24: During the experiment, the normal growth and development of cotton aphids remained uninterrupted. The data we gathered were derived from the continuous growth of these aphids. Based on our experimental findings, the reduction in the body length, width, and weight of cotton aphids can be attributed to alterations in their metabolic pathways following infection with Pseudomonas. These metabolic changes then impact the normal growth and development of A. gossypii.
Comments 25: Line 279: Here, Acyrthosiphon pisum is not the first time to state, so it should be written as “A. pisum”.
Response 25: Thank you for your comments We agreewith this comment. Therefore, according to the revision suggestions, we have changed “Acyrthosiphon pisum” to “A.pisum” on Line 330 of manuscript .
Comments 26: Line 370: “2009, Leaf-nosed bat, Journal, Issue,”, confusing.
Response 26: Thank you for your comments. We agreewith this comment. In the literature, there was an error in the citation format. This has been rectified at line 280 of the manuscript. Additionally, the incorrect primer format present has been eliminated. The new and correct citation format can be found at line 463.
Comments 27: Line 369: I have not seen the references formats like this. Authors are encouraged to revise the formats according to “Instructions for Authors”.
Response 27: Thank you for your comments. We agreewith this comment. In accordance with the "Instructions for Authors", we incorporated references formatted in the "APA 7th" style.

Reviewer 2 Report
Comments and Suggestions for Authors
Here, the authors investigated the relationship between Aphis gossypii and Pseudomonas. Aphis gossypii development, growth, and litter size was disrupted among aphids that were infected with Pseudomonas. Aphids that were infected with Pseudomonas also showed an upregulation in genes related to metabolism and energy synthesis, while a down regulation in genes with alkaline phosphate and histone proteinase B. This study provides novel and detailed information regarding A. gossypii that are infected with Psuedomonas, while effectively displaying their results through the use of figures. However, there are improvements that can be made. The abstract does not appear to be in the appropriate format required for Insects. I also did not see a simple summary. The authors should go back through their methods section to add additional details so that if someone were to want to repeat their experiments, they could – see specific comments regarding this below. The authors did a good in relating their work to the current literature, however, application of their results to other aphid species should be discussed more. It would also be valuable to discuss potential management applications of the authors’ findings and tie this to what has been previously done in similar systems and how it could be applicable to the system worked on in this study.
Overall, the findings of this study are valuable and novel. More details needs to be added to the methods to improve transparency and project repeatability, and more information should be added to the discussion to discuss how the authors finding can improve/contribute to A. gossypii management and how their work can be applied to other aphid species.
Specific comments:
Line 18: Introduce species name here with common name
Line 24 – Consider changing “litter size” to “offspring size”. I have not seen the term “litter size” used in the aphid literature before. Change throughout if decided.
Line 38 – add a space after “families” and the open parentheses. This occurs in several places in the paper. Correct throughout.
Line 43 – Do you mean insecticide resistance? Chemicals could include herbicides, fungicides, etc.
Line 45-46 – cite references to support this claim
Line 101 – change “A. gossypii” to “Aphis gopssypii”
Lines 116-117 – Say how this was measured
Line 126 – Is there product specific information that should be listed for Trizol?
Line 128 – how was integrity assessed with the gel electrophoresis?
Line 153 – correct citation font size
Lines 173-174 – did the fourth instar always come up the fourth day after birth, or is this an assumption? If assumption/not confirmed then delete “(fourth day after birth)”
Line 175 – what balance? This was not described earlier.
Line 189 – I do not see a Table 2 – add this in/correct wording accordingly
Lines 189-190 – is there a measurement of error you can add to these values – if so add it and state what measurement of error that is
Lines 191-192 – Add estimates, error, and significance values here as available
Line 322 – This statement can be applied to A. gossypii since that is what you tested in this study, but how confidently could it be applied to other aphid species?
Lines 353-363 – Reduce figure panel letter size by about 30-50% in Figures 1-3. They are too big now as is
Author Response
Comments and Suggestions for Authors
Here, the authors investigated the relationship between Aphis gossypii and Pseudomonas. Aphis gossypii development, growth, and offspring size was disrupted among aphids that were infected with Pseudomonas. Aphids that were infected with Pseudomonas also showed an upregulation in genes related to metabolism and energy synthesis, while a down regulation in genes with alkaline phosphate and histone proteinase B. This study provides novel and detailed information regarding A. gossypii that are infected with Psuedomonas, while effectively displaying their results through the use of figures. However, there are improvements that can be made. The abstract does not appear to be in the appropriate format required for Insects. I also did not see a simple summary. The authors should go back through their methods section to add additional details so that if someone were to want to repeat their experiments, they could – see specific Comments regarding this below. The authors did a good in relating their work to the current literature, however, application of their results to other aphid species should be discussed more. It would also be valuable to discuss potential management applications of the authors’ findings and tie this to what has been previously done in similar systems and how it could be applicable to the system worked on in this study.
Overall, the findings of this study are valuable and novel. More details needs to be added to the methods to improve transparency and project repeatability, and more information should be added to the discussion to discuss how the authors finding can improve/contribute to A. gossypii management and how their work can be applied to other aphid species.
Dear reviewer:
We are deeply grateful for your affirmation of our work. Your recognition, along with the valuable insights and suggestions you provided, is highly appreciated. In light of your feedback, we have meticulously reviewed and revised the manuscript. We have rectified the errors present and supplemented relevant content. Specifically, we have expanded the "Materials and Methods" section with more detailed information, ensuring a more precise account of our experimental approach.
Once again, we express our sincere gratitude for your constructive suggestions and comments regarding our manuscript.
Specific Comments:
Comments 1: Line 18: Introduce species name here with common name
Response 1: Thank you for your comments. We agreewith this comment. We have changed “cotton aphid” to “Aphis gossypii Glover” on Line 18.
Comments 2: Line 24 – Consider changing “offspring size” to “offspring size”. I have not seen the term “litter size” used in the aphid literature before. Change throughout if decided.
Response 2: Thank you for your comments.. We agreewith this comment. In the manuscript, we have replaced every instance of “little size” with “offspring size” in both the main text and the images.
Comments 3: Line 38 – add a space after “families” and the open parentheses. This occurs in several places in the paper. Correct throughout.
Response 3: Thank you for your comments. We agreewith this comment. Consequently, we have thoroughly examined and rectified similar issues across the entire text.
Comments 4: Line 43 – Do you mean insecticide resistance? Chemicals could include herbicides, fungicides,
Response 4: Thank you for your comments. We agreewith this comment. The intended meaning in this context pertains to the resistance of aphids to insecticide insecticides. In the article, the term “chemical resistance” has been amended to “Insecticide resistance” to convey this meaning accurately on Line 53-54.
Comments 5: Line 45-46 – cite references to support this claim
Response 5: Thank you for your comments We agreewith this comment. We have included two references at the corresponding locations to provide a scientific underpinning for our argument. These references not only support our stance but also enhance the credibility of our overall statement on Line 68-69.
Comments 6: Line 101 – change “A. gossypii” to “Aphis gopssypii”
Response 6: Thank you for your comments. We agreewith this comment. We have change “A. gossypii” to “Aphis gopssypii” on Line 107.
Comments 7: Lines 116-117 – Say how this was measured
Response 7: Thank you for your comments. According to the 16S sequencing results, obtain the gene sequence of Pseudomonas and utilize the NCBI Primer platform to design specific primers for Pseudomonas. Using absolute quantitative testing, we identified A. gossypii populations that were either free from infection or had minimal infection levels. These populations were designated as the control group (no-Pse). Separately, we picked A. gossypii populations infected with Pseudomonas to be the subjects of our experimental study (Pse).
Comments 8: Line 126 – Is there product specific information that should be listed for Trizol?
Response 8: Thank you for your comments. We have added comprehensive product-specific details.
Comments 9: Line 128 – how was integrity assessed with the gel electrophoresis?
Response 9: Thank you for your comments. The complete total RNA will produce clear 28S and 18S rRNA bands (eukaryotic samples) during denaturing gel electrophoresis. The intensity of the 28S rRNA band should be approximately twice that of the 18S rRNA band. This 2:1 ratio (28S: 18S) is a good indicator for assessing RNA integrity. Partially degraded RNA samples may exhibit diffuse bands in electrophoresis, without clear rRNA bands or the 2:1 ratio (28S: 18S) that only high-quality RNA possesses, as mentioned earlier. Completely degraded RNA will appear as dispersed at extremely low molecular weights.
Comments 10: Line 153 – correct citation font size
Response 10: Thank you for your comments. We agreewith this comment. In the manuscript, the superscript for (K.J. Livak and T.D. Schmittgen, 2001) has been removed on Line 168.
Comments 11: Lines 173-174 – did the fourth instar always come up the fourth day after birth, or is this an assumption? If assumption/not confirmed then delete “(fourth day after birth)”
Response 11: Thank you for your comments. Based on the data compilation from our laboratory, we've determined that the duration of each instar stage in cotton aphids is approximately 1day . Consequently, a cotton aphid is in its fourth - instar larval stage on the fourth day after its birth. Nevertheless, we plan to incorporate your suggestion in the manuscript and remove the phrase "fourth day after birth."
Comments 12: Line 175 – what balance? This was not described earlier.
Response 12: Thank you for your comments. This suggests that the daily fecundity (offspring size) of cotton aphids remains relatively consistent. In Figure 1, a trend graph depicting the fecundity (offspring size) of cotton aphids has been incorporated.
Comments 13: Line 189 – I do not see a Table 2 – add this in/correct wording accordingly
Response 13: Thank you for your commentst. Table 2 has been added to the Line 226 of manuscript.
Comments 14: Lines 189-190 – is there a measurement of error you can add to these values – if so add it and state what measurement of error that is
Response 14: Thank you for your comments. This section has been presented in the form of a line chart (Figure 1D). The line chart enables a clear visualization of the trend in the offspring size of A. gossypii.
Comments 15: Lines 191-192 – Add estimates, error, and significance values here as available
Response 15: Thank you for your comments. We agreewith this comment. This section has been added to Line 210.
Comments 16: Line 322 – This statement can be applied to A. gossypii since that is what you tested in this study, but how confidently could it be applied to other aphid species?
Response 16: Thank you for your comments. The aphids mentioned throughout this work specifically pertain to the cotton aphid species, Aphis gossypii. We have incorporated corresponding revisions within the manuscript to ensure this clarity on Line 373.
Comments 17: Lines 353-363 – Reduce figure panel letter size by about 30-50% in Figures 1-3. They are too big now as is
Response 17: Thank you for your comments. We agreewith this comment. We have carried out suitable modifications on Figures 1 - 4 to enhance their aesthetic appeal.

Round 2
Reviewer 1 Report
Comments and Suggestions for Authors
Through the manuscript has been revised, there are still some problems that need to be corrected. Based on these problems, I still doubt whether the authors reviewed the manuscript carefully before submission.
Line 21: “A. gossypii”.
Line 24-26: Confusing. This sentence should be revised carefully.
Line 129-138: These sentences could be merged into one paragraph.
Line 149: “A. gossypii”.
Line 182: “TWosex - Mschart”? Please note the use of capital and lowercase letters.
Line 268-269: "f, n, m”.
Line 278: “R0”.
Line 286-287: “Table 3”, it should start on a new line.
References: the scientific names should be italic.
Author Response
Comments and Suggestions for Authors
Through the manuscript has been revised, there are still some problems that need to be corrected. Based on these problems, I still doubt whether the authors reviewed the manuscript carefully before submission.
Dear reviewer:
We're extremely grateful for the significant comments and suggestions you provided regarding the manuscript. We have made the corresponding modifications in it.
Comments 1: Line 21: “A. gossypii”.
Response 1: Thank you for your comments. We agreewith this comment. We have revised "Aphis gossypii Glover" to "A. gossypii" on line 21 of the manuscript.
Comments 2: Line 24-26: Confusing. This sentence should be revised carefully.
Response 2: Thank you for your comments. We agreewith this comment.We have rewritten this sentence to make it easier to understand on line 24-26 of the manuscript.
Comments 3: Line 129-138: These sentences could be merged into one paragraph.
Response 3: Thank you for your comments. We agreewith this comment. We have adjusted these two paragraphs and integrated the rearing environment of the A. gossypii into lines 133-136 of the manuscript.
Comments 4: Line 149: “A. gossypii”.
Response 4: Thank you for your comments. We agreewith this comment. We have revised "aphis" to "A. gossypii" on line 151 of the manuscript.
Comments 5: Line 182: “TWosex - Mschart”? Please note the use of capital and lowercase letters.
Response 5: Thank you for your comments. We agreewith this comment. We corrected this error on line 145 of the manuscript.
Comments 6: Line 268-269: "f, n, m”.
Response 6: Thank you for your comments. We agreewith this comment.We have made corresponding modifications in lines 223-235 of the manuscript.
Comments 7: Line 278: “R0”.
Response 7: Thank you for your comments. We agreewith this comment. We have made corresponding modifications in Table 3 of the manuscript.
Comments 8: Line 286-287: “Table 3”, it should start on a new line.
Response 8: Thank you for your comments. We agreewith this comment. We have made adjustments to Table 3 in the manuscript.
References: the scientific names should be italic.
Response 9: Thank you for your comments. We agreewith this comment. We have italicized the scientific names of the references.
